# The Impact on Antioxidant Enzyme Activity and Related Gene Expression Following Adult Zebrafish (*Danio rerio*) Exposure to Dimethyl Phthalate

**DOI:** 10.3390/ani10040717

**Published:** 2020-04-20

**Authors:** Bailin Cong, Cong Liu, Lujie Wang, Yingmei Chai

**Affiliations:** 1The First Institute of Oceanography, Ministry of Natural Resources, Qingdao 266061, China; biolin@fio.org.cn; 2Department of Marine Science, Marine College, Shandong University (Weihai), Weihai 264209, China; excong@163.com (C.L.); toodloo@126.com (L.W.)

**Keywords:** zebrafish, DMP, antioxidant enzyme activity, gene expression

## Abstract

**Simple Summary:**

Dimethyl phthalate (DMP) is a widespread environmental contaminant and its toxicological effects on fish have not been adequately examined. Our present study clearly showed that a high concentration induced oxidative damage in zebrafish, which proved the molecular regulation due to the negative effects of DMP, along with the physical damage in zebrafish. We also found that antioxidant enzymes might be used as appropriate biochemical markers for the toxic identification of DMP.

**Abstract:**

Dimethyl phthalate (DMP) is a widespread environmental contaminant that poses potential toxicity risks for animals and humans. However, the toxicological effects of DMP on fish have not been adequately examined. In this study, the acute toxicity, oxidative damage, antioxidant enzyme activities, and relative gene expression patterns were investigated in the liver of adult zebrafish (*Danio rerio*) exposed to DMP. We found that the lethal concentration (LC_50_) of DMP for zebrafish after 96 h of exposure was 45.8 mg/L. The zebrafish that were exposed to low, medium and high concentrations of DMP (0.5, 4.6, and 22.9 mg/L, respectively) for 96 h had an increased malondialdehyde (MDA) content and a lower antioxidant capacity compared with the control solvent group. The total superoxide dismutase (SOD) activity was significantly higher than 0 h after initial exposure for 24 h at low concentrations, and then decreased at high concentrations after exposure for 96 h. The catalase (CAT) and glutathione S-transferase (GST) activities were significantly reduced after 96 h of exposure to high concentrations of DMP, with the up- or down-regulation of the related transcriptional expression. These findings indicated that DMP could cause physiological effects in zebrafish by disturbing the expression levels of antioxidant enzymes. These results might contribute to the identification of biomarkers to monitor phthalate pollution.

## 1. Introduction

Phthalic acid esters (PAEs) are a class of ubiquitous chemicals in the environment, widely used as plasticizers for plastics in many household and industrial products [1,2,3]. Low molecular weight dimethyl phthalate (DMP) is one of the most common and extensively used PAEs that has been frequently reported in various environmental samples, including marine water, freshwater, sediment, soil, fatty foods and cosmetics, and the concentrations of DMP in landfill leachate has reached 300 mg/L [4,5]. The frequency of DMP contamination has been detected to be 37% in common food samples from a market in Albany, New York, and significantly higher in China [6,7]. Due to its wide applications in many kinds of products, such as perfumes, paints, waxes, pharmaceutical products, insecticide materials, adhesives, printing inks, coatings and cosmetics, DMP has been designated as an environmental priority organic pollutant by many regulators, such as the United States, the European Union and China [8,9]. Because of its simplicity and relative stability under natural environmental conditions, its half-life is approximately 3 years, and its highest solubility in water is 4500 mg/L at 25 °C. DMP has been recognized as posing a potential threat to the environment and human health [10]. The toxicological properties of PAEs include the hepatotoxic, reproductive/developmental, endocrine, and neurological effects in humans and animals [11,12,13,14], and the acute exposure via inhalation leads to irritation of the eyes, nose, and throat in humans and animals [15,16]. Short- to intermediate-term exposure induces decrements in body weight gain, changes in hemoglobin and increases in absolute and relative liver weights [17].

Based on standard ecotoxicity tests, employing more detectable responses to the expression levels of antioxidant enzyme genes and enzymatic activities may be used as reliable biochemical markers for the toxicity evaluation of PAE pollutants [18,19,20,21]. The increasing of oxidative stress can be induced by a number of phthalates via influencing antioxidant enzyme activities in some organisms, including fish [18,19,20,21,22]. Zhang et al. (2014) discovered that low concentrations of benzyl butyl phthalate (BBP) could stimulate superoxide dismutase (SOD) activity, while high concentrations inhibited this activity and the mRNA levels of SOD inhibition decreased and then increased in the muscles of zebrafish after 7 days of exposure [23]. Yang et al. (2018) demonstrated that di (2-ethylhexyl) phthalate (DEHP) exposure at 20, 100 and 200 µg/L over 21 days suppressed the activity and transcriptional expression of the antioxidant enzymes SOD, catalase (CAT) and glutathione s-transferase (GST) in the fry of medaka fish [24]. More recently, Qu et al. (2015) and Zheng et al. (2013) reported that in *Carassius auratus*, when the fish were injected with 17 different PAEs at a concentration of 10 mg/kg for several days, respectively, the activities of antioxidant enzymes (SOD and CAT) were inhibited by DMP with an increasing treatment time [19,25]. These findings suggest that subacute concentrations of PAEs, including DMP, can induce oxidative stress in fish.

To date, research on the toxicological effects of DMP in humans and aquatic animals is still limited. Zebrafish are commonly used as a promising model organism for studies on the toxicity of pollutants due to their high homology with vertebrates [26,27]. The present study investigated the effects of DMP on the amount of malondialdehyde (MDA), antioxidant enzymatic activities and mRNA levels in the livers of zebrafish in batch mode under controlled conditions. This information is intended to find useful biomarkers of DMP pollution.

## 2. Materials and Methods

### 2.1. Experimental Materials

Four-month-old zebrafish (a total of 500) were obtained from a local fish dealer (Weihai, China) and reared in 500 L glass tanks (containing 400 L of dechlorinated and charcoal-filtered tap water) for 2 weeks at 25 ± 1 °C, pH 7.2 ± 0.3, and a dissolved oxygen content of 6.00 ppm with 14 h light/10 h dark cycles. The fish were fed 2 times a day with tropical flake food. Before DMP exposure, 20 fish were acclimated in 20 L glass tanks for 1 week. All experiments were performed under the approval of the Committee on Animal Care and Use and the Committee on the Ethics of Animal Experiments of Shandong University at Weihai. All of the zebrafish collections and the anatomy experiments were conducted in accordance with the “Guidelines for Experimental Animals” of the Ministry of Science and Technology (Beijing, China; No. [2006] 398, 30 September 2006). Zebrafish hypothermic shock was performed in an ice bath (5 parts ice to 1 part system water at a constant temperature of 2–4 °C) [28].

DMP (CAS No. 131-11-3) was purchased from the Beijing Solarbio Science and Technology Co. Ltd. (Beijing, China). DMP for the exposure experiments was solubilized in acetone and the chemicals used were of analytical grade [29].

### 2.2. Exposure Experiments and Sample Collection

A toxicity test to assess the 96 h LC_50_ value of DMP in zebrafish was performed in a laboratory system according to Organisation for Economic Co-operation and Development (OECD) guidelines [30]. The initial concentrations of DMP used were 25, 50, 100, 150, and 200 mg/L. Ten zebrafish in each experimental tank (20 L) were introduced for 96 h based on a range-finding test prior to the experiment. The blank control and solvent control received 0.004% acetone (v/v), the same concentration used in the test groups which contained a series of concentrations (25, 50, 100, 150, and 200 mg/L DMP) [31]. For each treatment, 5 replicates with 10 fish each were conducted (blank control, solvent control, plus DMP). The test fish were not fed, and the behavioral and morphological features were checked by visual analyses.

On the basis of the zebrafish mortality in each tank, acute toxicity was expressed as the LC_50_ of DMP, which was calculated by using the probit analysis method [32].

After the LC_50_ value was determined, the zebrafish were exposed to subacute toxic concentrations of DMP (0.5, 4.6, and 22.9 mg/L), and parallel blank control and solvent control groups were used in the exposure experiments. Each concentration in the DMP groups had 4 replicates with 80 fish each, and the control groups had the same replicates. For each time endpoint (24, 48, and 96 h), liver samples were dissected from 20 living fish (2.5 ± 0.3 cm and weight 0.22 ± 0.05 g) in each group via a stereomicroscope washing twice with phosphate buffer saline (PBS) on an ice-cold plate and divided into 2 samples: 15 individuals were used to assay MDA and antioxidant enzyme activity, and 5 individuals were used for real-time polymerase chain reaction (RT-PCR).

### 2.3. Biochemical Assays

Each liver sample was homogenized in a homogenizer in ice-cold buffer (0.1 M Tris-HCl, 0.1 mM-EDTA, 0.1% Triton X-100 (v/v), pH 7.8). The homogenates were centrifuged, and the supernatants were collected. The supernatants were used for biochemical parameter measurements. The protein content, MDA, total antioxidant capacity (T-AOC), and the activities of the antioxidant enzymes were analyzed using kits (Nanjing Jiancheng Bioengineering Institute, Nanjing, China) according to the manufacturer’s instructions. The quantity of protein was measured by the Bradford method using bovine serum albumin as a standard [33]. The T-AOC was detected by the reducing ability of ferric ions to ferrous ions and expressed as U/mg protein (nmol of ferric ions reduced per min/mg protein). The MDA levels were measured by the conversion of thiobarbituric acid to reactive substances, which had a high absorbance at 532 nm. The MDA concentrations were expressed as nmol/mg protein.

The total SOD activity was measured by the nitro blue tetrazolium (NBT) method at 560 nm [34]. The enzymatic activity was calculated as U/mg protein. One unit of SOD was defined as the amount of sample required to inhibit the rate of reduction of NBT by 50%. The CAT activity was detected by the decrease in hydrogen peroxide radical (H_2_O_2_) concentration at 240 nm [35]. The CAT activity (1 U) was expressed as 1 mmol of decomposed hydrogen peroxide per second per mg of protein. The glutathione S-transferase (GST) activity was determined by 1-chloro-2, 4-dinitrobenzene colorimetry, and the absorbance was measured at 340 nm for 5 min. The GST activity was expressed as the number of μmol in 1 minute per mg of protein (μmol/mg protein/min).

### 2.4. Molecular Studies

The total RNA was extracted from the liver tissues in each treatment group by using TRIzol reagent according to the manufacturer’s protocol (Invitrogen, Carlsbad, CA, USA). First-strand cDNA was synthesized by the use of the SMART cDNA method (BD Biosciences-Clontech, Palo Alto, CA, USA). Finally, quantitative RT-PCR (RT-PCR) was performed on a 7300 real-time system (Applied Biosystems, Foster City, CA, USA). The PCR mix (Takara, TB Green™ Premix Ex Taq II, Chiryu Shi, Japan) was denatured at 94 °C for 3 min, followed by 40 cycles of amplification (94 °C for 15 s, 58 °C for 20 s, and 72 °C for 20 s). All reactions were repeated in triplicate for the RT-PCR analysis. *Beta-actin* was used as the housekeeping gene, and the changes in the abundances of transcripts of each target gene were expressed as its ratio to the expression of the reference gene (Table 1). A significant difference was accepted at *p* < 0.05. The reaction of each template was performed in triplicate. The relative gene expression level was calculated using the comparative Ct (2-ΔΔCt) method [36]. The types of tested *SOD*, *CAT* and *GST* gene were *superoxide dismutase 1*, *catalase transcript variant X2* and *glutathione S-transferase pi 1*, respectively.

### 2.5. Statistical Analyses

The statistical analyses were performed with SPSS 19.0 and Excel 2007 software. The experimental data were expressed as the mean ± standard deviation. Student’s *t*-test was used to examine the differences between the treatment groups and control groups (the significance level was *p* < 0.05).

## 3. Results

### 3.1. Effects of DMP on Zebrafish Survival

The mortality of the zebrafish treated with different concentrations of DMP was seen to increase in a dose-dependent manner (Figure 1). Figure 1 shows survivorship curves for DMP treatment in the zebrafish at various concentrations in water after 96 h of exposure. There was only 20% mortality in the 25 mg/L group within 96 h, and the remaining fish survived. In the 100 mg/L DMP-treated group, there was 80% mortality in 96 h. There was 100% mortality in the 200 mg/L group, and mortality occurred within 96 h. Therefore, the LC_50_ value of DMP for zebrafish was calculated to be 45.8 mg/L. The zebrafish (>80%) exposed in the experimental tanks containing 100 to 200 mg/L showed less movement and body shape changes compared to the solvent control after 24 h of incubation. After 72 h of incubation, >90% of these zebrafish floated on the surface of the water and died. More than 90% of the zebrafish died within 48 h of incubation in 200 mg/L DMP. In contrast, all the control zebrafish survived.

### 3.2. Effect of DMP on Oxidative Stress in Zebrafish

To determine the effects of DMP on lipid peroxidation, the zebrafish were exposed to three sublethal concentrations of DMP (0.5, 4. 6, and 22.9 mg/L) for 24, 48 and 96 h (Figure 2). As shown in Figure 2, the MDA content was elevated in the low concentration experimental group at 24 h, levelled off after 48 h (*p* < 0.05, Figure 2), and increased in the medium and high concentration treatment groups at all time points.

As shown in Figure 3, the T-AOC values of fish after low concentrations of DMP treatment increased significantly at 24 h and then gradually decreased. However, the fish exposed to high concentrations of DMP showed an obvious reduction in the T-AOC values.

The activity levels of the primary antioxidant enzymes SOD, CAT, and GST after exposure to DMP are shown in Figure 4a–c. Upon treatment with a low concentration of DMP for 24 h, there was an increased level in the SOD activity and no obvious changes in the levels of CAT and GST activity; after 48 h, the SOD activity was reduced and the GST activity increased significantly. The treatment of fish with high concentrations of DMP showed a decline in SOD and CAT activities at 24 h, and GST activity increased. After 96 h of exposure to high concentrations of DMP, the activities of SOD, CAT, and GST all decreased gradually with time and were lower than the control groups (Figure 4a–c).

### 3.3. Effects of DMP on the Transcription of Related Antioxidant Enzymes

The gene expression levels of the antioxidant enzymes, *SOD*, *CAT* and *GST* after DMP exposure were investigated (Figure 5a–c). The fish exposed to low, medium, and high concentrations of DMP for 24, 48, and 96 h showed significant up- or down-regulation in the gene expression levels of *SOD*, *CAT* and *GST*. The gene expression of *CAT* after 48 h of DMP treatment was reduced significantly. Low concentrations of DMP showed no obvious change in *CAT* mRNA levels after 24 h, but *SOD* and *GST* transcription increased notably after 48 h. Medium and high concentrations of DMP resulted in the up-regulation of *SOD* mRNA expression after 24 and 48 h followed by down-regulation of *SOD* and *GST* expression after 96 h.

## 4. Discussion

PAEs have become global pollutants that contaminate water resources and affect aquatic organism health, including fish. The present study investigated the toxic effects of a low molecular weight phthalate, DMP, on zebrafish. In the present study, the 96 h LC_50_ of DMP was found to be 45.8 mg/L in zebrafish. Similar results were obtained in a study of other fish species; the LC_50_ of DMP has been reported to be 50 mg/L for bluegills (*Lepomis macrochirus*), 55.8 mg/L for black molly (*Poecilia sphenops*), 56 mg/L for rainbow trout (*Oncorhynchus mykiss*), and 39 mg/L (flow through) and 121 mg/L (static) for fathead minnows (*Pimephales promelas*) [37,38,39]. Moreover, zebrafish embryos develop a low incidence of general dysmorphology at 1 × 10^−5^ or 1 × 10^−4^ M DMP [40] The LC_50_ of DMP varies between 0.48 and 121 mg/L in aquatic fish species [41,42]. This is attributed to the experimental conditions and several other factors, such as different fish species, age, feeding habits, and sex.

Under the stress of exogenous factors (such as chemical contaminants), organisms will produce a lot of harmful free radicals, like superoxide radicals, hydrogen peroxide radicals and hydroxyl radicals. The accumulation of free radicals stimulates the organism to play the role of the antioxidant defense system to avoid biological damage [43]. However, when the generation rate of free radicals is faster than its elimination, organisms will be oxidatively damaged, including enzymes inactivation, DNA and cholesterol damage, and the peroxidation of unsaturated fats in the cell membrane. It will destroy the integrity of the cell membrane caused by undergoing lipid peroxidation [44]. Therefore, antioxidant defense is an important defense against the impact of environmental contaminants. The involvement of antioxidant enzymes (such as SOD, CAT, and GST) can protect cells from oxidant damage. Among antioxidant enzymes, SOD plays an important role in the first defense system that is involved in the detoxification of reactive oxygen free radicals and H_2_O_2_ [45]. CAT, another key antioxidant defense enzyme, is regarded as an important biomarker to estimate oxidative stress via alterations in activity, and GST plays a key role in detoxifying a variety of electrophilic substances into hydrophilic compounds as a second-phase detoxification enzyme and antioxidant [46,47,48]. MDA is a final important product of lipid peroxidation from oxidative stress, and analyzing its production can reflect damage to organisms [49]. As such, the degree of lipid peroxidation in an animal can be indirectly measured by MDA. In the present study, the MDA content in zebrafish showed an increasing trend at 24 h and a decreasing trend at 96 h upon exposure to low concentrations of DMP, and with high concentrations of DMP and prolonged exposure, MDA accumulated. This indicated that the antioxidant mechanism of the fish treated with low concentrations of DMP for 24 h can effectively eliminate oxidative damage. However, after 96 h of exposure to high concentrations of DMP, the oxidative damage induced by DMP became severe in fish.

T-AOC level is a comprehensive indicator to measure the function of the antioxidant systems. In this experiment, the highest activity of T-AOC enzyme was tested at a low concentration of DMP for 24 h, while the lowest activity was tested at a high concentration for 96 h. This suggests that zebrafish liver tissue has the highest antioxidant capacity under low concentration (24 h) DMP stress, but under high concentration DMP stress, the body is oxidatively damaged and the total antioxidant capacity decreases. Zhang et al. also reported that the highest activity of T-AOC in the liver of *Carassius auratus* was detected when exposed to B(a)P. As the B(a)P concentration increases, T-AOC activity in the liver gradually decreases [50].

We found that the activity of SOD presented a rising trend at 24 h and a falling trend at 96 h. This might be due to the increasing production of oxygen free radicals that correspondingly leads to increasing SOD activity under the stress of DMP. CAT activity was reduced at high concentrations due to excessive amounts of DMP enhancing the stress, which leads to the depletion of the CAT and SOD enzymes. Similarly, 48 h of exposure to low concentrations of DMP led to increased GST activity, which was later reduced at 96 h. In a recent study, the effects of endocrine disrupting compounds on antioxidative balance were investigated in wildlife vertebrates [51]. Similar results were also observed in the livers of *Carassius auratus* injected intraperitoneally with 17 different phthalates at a concentration of 10 mg/kg for 10 days [25]. Many previous studies have shown that oxidative stress status is interrelated with pollutant exposure, which is a possible regulation of the toxicity of these chemicals [52,53,54,55,56,57,58,59].

In this study, SOD, CAT and GST enzyme activities in zebrafish liver were consistent with T-AOC, and the highest expression was detected at low concentrations of DMP contrast in the lowest under high concentrations of DMP stress. This shows that under the stress of DMP, the change pattern of antioxidant enzymes in zebrafish is consistent with its own antioxidant capacity, which indicates the level of environmental pollution and oxidative stress in the zebrafish and the key role of SOD, CAT and GST in the antioxidant defense system.

Changes in transcriptional levels are commonly used as the earliest and most sensitive biomarkers for physiological responses to environmental stress. Our results showed that DMP exposure down-regulated the expression levels of *CAT* and *GST*. These results are consistent with the enzymatic activity and T-AOC levels after exposure to DMP. Irrespective of the mechanism in zebrafish, the toxicity of DMP may lead to lower antioxidant enzyme activity and cause oxidative damage to the organism.

## 5. Conclusions

The present study clearly showed that a high concentration of DMP induced oxidative damage in zebrafish. We found that high or low antioxidant enzyme activity could affect the survival and health status of zebrafish and consequently cause increased fish mortality. This study also determined that DMP disturbed the gene expression levels of *SOD*, *CAT*, and *GST*; therefore, antioxidant enzymes might be used as appropriate biochemical markers for the toxic identification of DMP. Further studies are needed to elucidate whether exposure to plasticizers and alterations to particular isoforms of antioxidant enzymes are underlying factors of abnormality and malfunction in different systems of fish under long processes.

## Figures and Tables

**Figure 1 animals-10-00717-f001:**
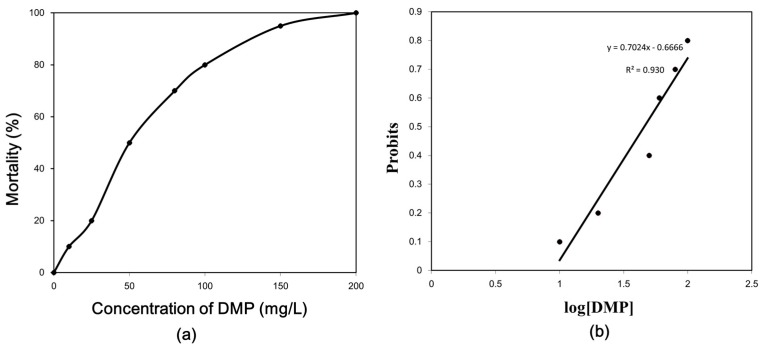
Dose–response relationships after 96 h of exposure for dimethyl phthalate (DMP): (**a**) mortality of zebrafish treated with different concentrations of DMP; (**b**) probit models at different concentrations of DMP for calculation of the LC_50_.

**Figure 2 animals-10-00717-f002:**
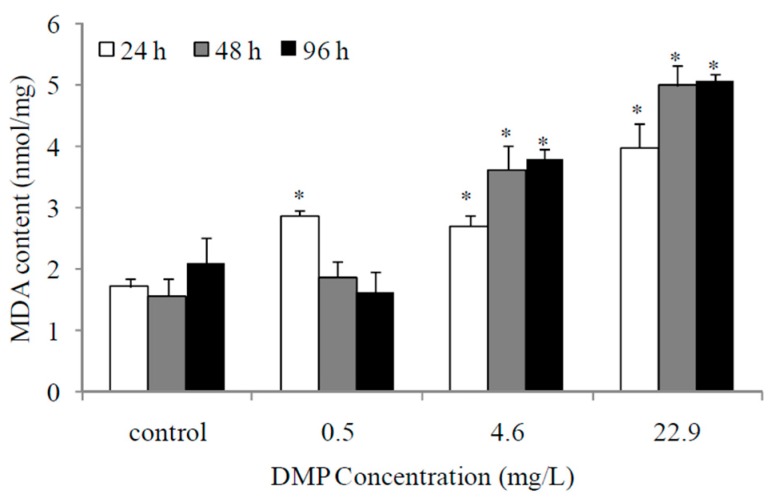
DMP-induced lipid peroxidation levels in zebrafish. Adult fish were exposed to 0.5, 4.6 and 22.9 mg/L DMP for 24, 48, and 96 h. Error bars represent the standard deviation. Significant differences from the control are indicated by an asterisk, which is based on Student’s *t*-test (*p* < 0.05).

**Figure 3 animals-10-00717-f003:**
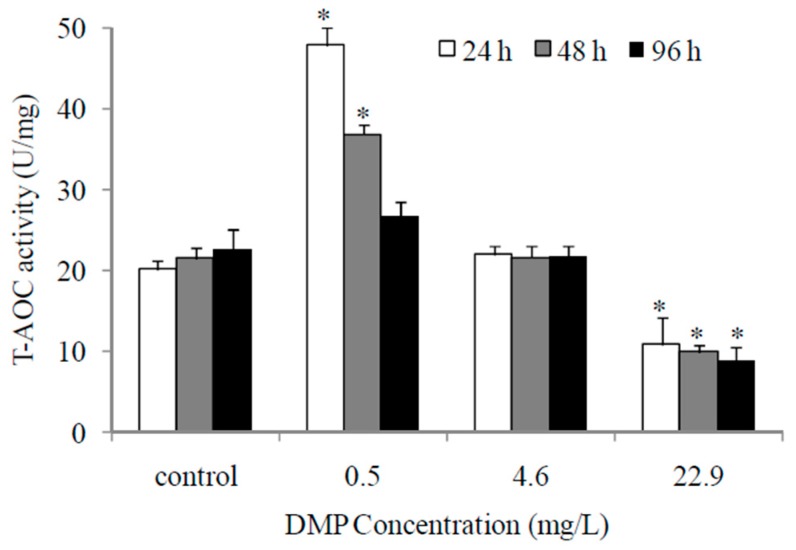
DMP-induced changes of total anti-occident capacity (T-AOC) in zebrafish. Adult fish were exposed to 0.5, 4.6 and 22.9 mg/L DMP for 24, 48, and 96 h. Statistical significance between experimental and control groups are indicated with asterisks: significance is based on Student’s *t*-test (*p* < 0.05). The error bars represent the standard deviation (n = 5).

**Figure 4 animals-10-00717-f004:**
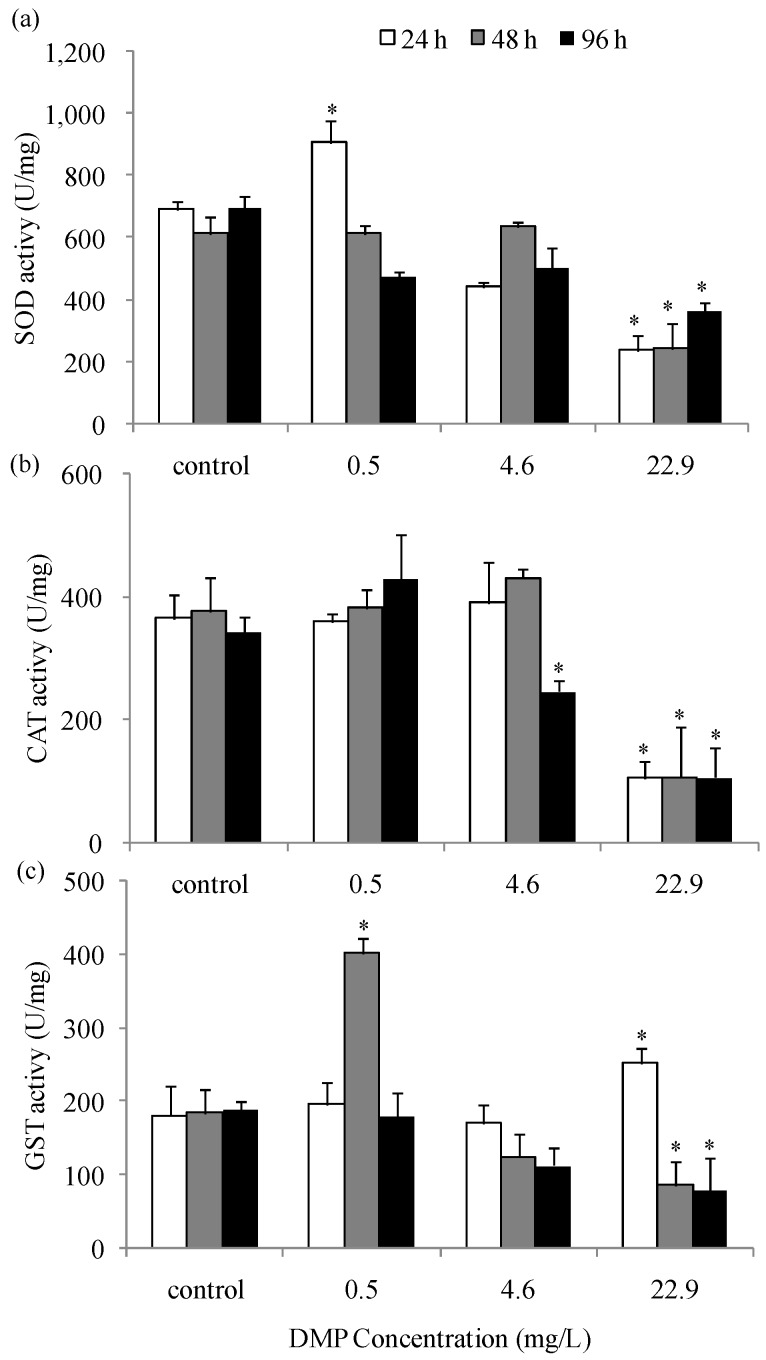
DMP-induced changes of antioxidant enzyme activity in zebrafish liver: (**a**) superoxide dismutase (SOD) activity; (**b**) catalse (CAT) activity; (**c**) glutathione s-transferese (GST) activity. Adult fish were exposed to 0.5, 4.6 and 22.9 mg/L DMP for 24, 48, and 96 h. Statistical significance between the experimental and control groups is indicated with an asterisk: *p* < 0.05. The error bars represent the standard deviation (n = 5).

**Figure 5 animals-10-00717-f005:**
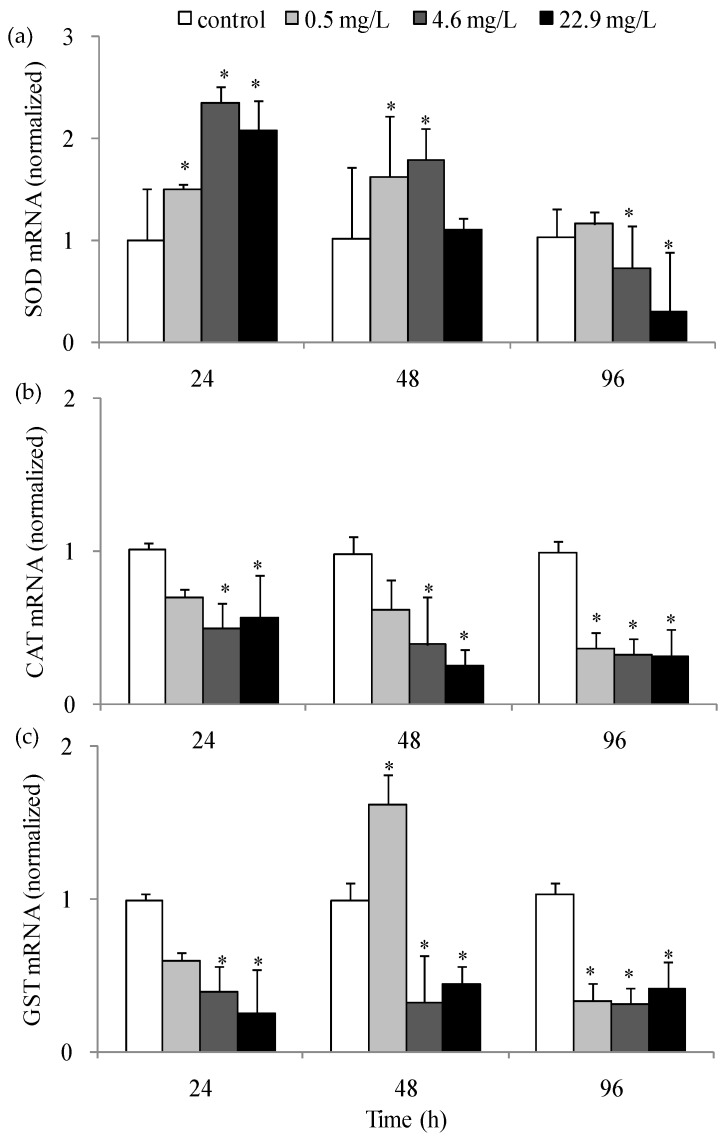
Gene expression level of antioxidant enzyme in zebrafish liver exposed to DMP: (**a**) the *SOD* gene expression; (**b**) *CAT* gene expression; (**c**) *GST* gene expression. Adult fish were treated with 0.5, 4.6 and 22.9 mg/L DMP for 24, 48, and 96 h. Gene expression levels of significance compared to the control are marked with asterisks (*p* < 0.05). The error bars represent the standard deviation (n = 5).

**Table 1 animals-10-00717-t001:** Primer sequences used for real-time-polymerase chain reaction (PCR).

Primer	Primer Sequence (5′-3′)	Annealing Temperature (Tm)	Product Length (bp)	Amplification Efficiency
*superoxide dismutase* (*SOD*) primer-F	TCCGCACTTCAACCCTCA	58.44	215	97.0%
*SOD* primer-R	CCTCATTGCCACCCTTCC	57.66		
*catalase* (*CAT*) primer-F	TACCAGTCAACTGCCCGTAC	59.40	145	96.5%
*CAT* primer-R	GACTCAAGGAAGCGTGGC	58.43		
*glutathione S-transferase* (*GST*) primer-F	CCAACCACCTCAAATGCT	55.09	150	98.1%
*GST* primer-R	ACGGGAAAGAGTCCAGACAG	59.03		
*Beta-actin* primer-F	CGAGCAGGAGATGGGAACC	59.86	214	99.5%
*Beta-actin* primer-R	CAACGGAAACGCTCATTGC	58.29		

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
