# Peer review of "The Impact on Antioxidant Enzyme Activity and Related Gene Expression Following Adult Zebrafish (Danio rerio) Exposure to Dimethyl Phthalate"

_animals, 2020, doi:10.3390/ani10040717_

Round 1
Reviewer 1 Report
Dear authors,
the manuscript has been modified according to previous recommendations
Author Response
Dear Reviewer
I would like to express my most sincere thanks for your comments which is really valuable for my manuscript and research.
Best wishes
Yimei Chai
Reviewer 2 Report
Regarding revision of paper “The impact on antioxidant enzyme activity and related gene expression following adult zebrafish (Danio rerio) exposure to dimethyl phthalate” by Cong et al. Manuscript. I think authors have done a really deep revision following given suggestions and now the manuscript seems to be really improved.
I propose just some minor corrections.
Molecular study paragraph:
Please specify which genes are you studying (SOD1, GSTp1… etc).
General comments:
Bibliography is still not in compliance with journal style, particularly, journal names are not in abbreviated form and some name of species are still not in italics
Point by-point:
Pag. 1 Line 12: “and its toxicological effects of DMP” please delete “of DMP”.
Pag. 1 Line 15: please delete “the model species of”
Pag. 1 Line 20: “in the livers” delete “s”.
Pag. 1 Line 21: correct “(LC50)” with “(LC50).
Pag. 2 Line 85: “In addition, zebrafish is not an endangered or protected species in China nor in other countries.” This is not necessary.
Pag. 2 Line 85: “Hypothermic shock of Zebrafish” change with “Zebrafish hypothermic shock”.
Pag. 4 Line 180-187: remove italics from acronyms and check that all of them are in uppercase. It is not necessary to add the p value in brackets.
Pag. 9: Pay attention with layout, something is wrong. Delete journal heading and check page numbers
Pag. 10 Line 282: “Carassius auratus”
Pag. 10 Line 303: “cat and gst” please uppercase.
Author Response
Dear Reviewer
Thanks a lot for your detailed comments. My responses are as following:
Please specify which genes are you studying (SOD1, GSTp1… etc).
These genes have been clarified in 2.4.
Bibliography is still not in compliance with journal style, particularly, journal names are not in abbreviated form and some name of species are still not in italics
These have been revised.
Pag. 1 Line 12: “and its toxicological effects of DMP” please delete “of DMP”.
It has been revised.
Pag. 1 Line 15: please delete “the model species of”
It has been revised.
Pag. 1 Line 20: “in the livers” delete “s”.
It has been revised.
Pag. 1 Line 21: correct “(LC50)” with “(LC50).
It has been revised.
Pag. 2 Line 85: “In addition, zebrafish is not an endangered or protected species in China nor in other countries.” This is not necessary.
It has been deleted.
Pag. 2 Line 85: “Hypothermic shock of Zebrafish” change with “Zebrafish hypothermic shock”.
It has been revised.
Pag. 4 Line 180-187: remove italics from acronyms and check that all of them are in uppercase. It is not necessary to add the p value in brackets.
It has been revised.
Pag. 9: Pay attention with layout, something is wrong. Delete journal heading and check page numbers
It has been revised.
Pag. 10 Line 282: “Carassius auratus”
It has been revised in line 280
Pag. 10 Line 303: “cat and gst” please uppercase.
It has been revised.
Best wishes
Yingmei Chai
Reviewer 3 Report
The revised version of the manuscript (R1) has been greatly improved over the original version. The manuscript can now be accepted for publication.
Author Response
Dear Reviewer
I would like to express my most sincere thanks for your comments which is really valuable for my manuscript and research.
Best wishes
Yimei Chai
This manuscript is a resubmission of an earlier submission. The following is a list of the peer review reports and author responses from that submission.
Round 1
Reviewer 1 Report
Dear authors,
I hope this review and message find you well.
Sincerely
The paper "The impact on antioxidant enzyme activity and related gene expression following adult zebrafish (Danio rerio) exposure to dimethyl phthalate" is generally well written and the topic is very interesting but lacks a few detailed information.
Title
-Line 4: Species name in the title is not in Italics
Abstract
-Line 21: Species name is not in Italics
Introduction
- Line 66: Species name is not in Italics
- Lines 72-73: “Zebrafish are commonly used as a promising model organism for studies on the toxicity of pollutants due to their high homology with vertebrates”. Please add some citations to support this concept. I would suggest:
- Pecoraro, R., Marino, F., Salvaggio, A., Capparucci, F., Di Caro, G., Iaria, C., ... & Scalisi, E. M. (2017). Evaluation of chronic nanosilver toxicity to adult zebrafish. Frontiers in physiology, 8, 1011.
- Pecoraro, R., D'Angelo, D., Filice, S., Scalese, S., Capparucci, F., Marino, F., ... & Salvaggio, A. (2018). Toxicity evaluation of graphene oxide and titania loaded nafion membranes in zebrafish. Frontiers in physiology, 8, 1039.
Materials and Methods
-Line 102: Please specify in materials and methods all procedures used to reduce pain and sufferance in fish. In particular specify methods for anaesthesia and euthanasia and support them with bibliography, I would suggest these two in which methods are fully described:
- Iaria, C., Saoca, C., Guerrera, M. C., Ciulli, S., Brundo, M. V., Piccione, G., & Lanteri, G. (2019). Occurrence of diseases in fish used for experimental research. Laboratory animals, 53(6), 619-629.
- Iaria, C., Migliore, S., Macri, D., Bivona, M., Capparucci, F., Gaglio, G., & Marino, F. (2019). Evidence of Centrocestus formosanus (Nishigori, 1924) in Zebrafish (Danio rerio). Zebrafish, 16(6), 522-526.
Results
-3.1.1., lines 152-156- Describe the results in a better order and always the same for concentrations and time points.
And: “Zebrafish (>80%) exposed in the experimental tanks containing 100 to 200 mg/L showed less movement and body shape changes compared to the control after 24 h of incubation. After 72 h of incubation, >90% of these zebrafish floated on the surface of the water and died. More than 90% of the zebrafish died within 48 h of incubation in 200 mg/L DMP”, what about 25 mg/L group?
-3.1.2., lines 160-162- Were significant differences found only for the 0.5 mg / L group after 48 hours of exposure? For the other groups, was p>0.05 at all time points? Please specify.
Lines 163-165- Add p value
Line 167- Replace “A, B and C” with a, b and c
-3.1.3., Line 176 Remove “)”; Add p value
-3.2.,
Fig1: please provide better fig.
Fig2: Asterisk not indicated. Add it
Fig3: Again, asterisk not indicated, please add it
Line 214: Replace “sod”, “cat” and “gst” with “sod”, “cat” and “gst”
Line 216: Delete “(c)”
Discussion
The discussion is generally poor and merely descriptive. The results obtained, their possible implications should be discussed more. Add more information about damage induced by DMP in fish.
Author Response
Dear Reviewer
Thank you so much for your comments which are very valuable for my manuscript and research. Please see the attachment of my responses.
Best wishes
Chai Yingmei

Reviewer 2 Report
Manuscript ID: Animals-709093
Regarding revision of paper “The impact on antioxidant enzyme activity and related gene expression following adult zebrafish (Danio rerio) exposure to dimethyl phthalate” by Cong et al. Manuscript ID: Animals-709093, the present study provides really interesting data about antioxidant activity of dimethyl phthalate in zebrafish. The topic is of great interest for researchers working in the field. However, there are several mistakes and doubts to be clarified.
In the belief that this manuscript would be of interest and usefulness to the research community, I suggest publication after a major revision has been undertaken.
General comments:
- Materials and methods are often confused, not detailed, and do not always assure reproducibility, especially on description of experimental groups and molecular analysis.
- Molecular studies description should be more detailed:
- How did authors check for quantity, purity and integrity of total RNA?
- What controls were used for molecular analysis?
- Could you provide information on total reaction volumes during all steps?
- How did the authors verify the absence of genomic DNA in total RNA?
- How did the authors study genes and primer? Were they reported in the Bibliography or did the authors design them?
- Table 1:
- In zebrafish there are different SOD genes (SOD1, SOD2, SOD3b and SOD3a) which one was investigated?
- Similarly, for GST (GSTa.1, GSTa.2, GSTp1, GSTr, GSTm.3… etc.)
- How did the authors choose the reference gene and why Beta-actin? Have there been any other reference genes tested before to use Beta-actin in statistical analysis?
- From which company were the primers obtained? And how were they tested prior to use? Could you provide information on amplicon size and efficiency?
- Regarding RT-PCR, which master mix was used?
- Did the authors perform any melting curve analysis?
- Can the authors provide any bibliography in support of these methods?
- Statistical analysis: How did the authors compare Relative quantity (RQ) values and normalize data?
- Figure: Fig 1, 2 and 3 do not show any significant differences or are asterisks missing?
- Bibliography must be checked and corrected, it does not conform with journal style, particularly, journal names are not in abbreviated form and name of species are not in italics. Moreover, years of publication are often missing throughout the text.
- Several species names are not in italics throughout the text and bibliography
- Low plagiarism has been detected (less than 19%) using Plagiarism Checker X software.
- The English is acceptable for general comprehension but several mistakes can be found throughout the text. A general revision by a native English speaker must be undertaken before publication.
Revision point-by-point:
Line 4: The species name is not in italics
Line 14: Please change “molecular regulations of DMP negative effect and physical damage” to “molecular regulation due to the negative effects of DMP, along with physical damage”
Line 21: Species name not in italics
Line 25: Please correct in “with the control group”
Line 25: “was significantly higher” higher than what? than normal? Than control? Than zero hour?
Line 27: Please change “The activities of catalase (CAT) and glutathione S-transferase (GST) were significantly reduced with 96 h” to “Catalase (CAT) and glutathione S-transferase (GST) activities were significantly reduced after 96 h of exposure”
Line 37: Please change “plastic plasticizers” with “plasticizers for plastics”
Line 45: Better to change “by related governments” with “by many regulators”
Line 55-57: This sentence is not clear, please rewrite and clarify
Line 59: Please add the year to citation “Zhang et al.”
Line 59: Please correct with “Benzyl butyl phthalate (BBP)”
Line 61: “mRNA levels during SOD inhibition decreased” it is not clear when decrease occurred
Line 62: The year is missing in “Yang et al.”
Line 63: Please correct with Di(2-ethylhexyl)Phthalate (DEHP)
Line 65: Add the year to “Qu et al.” and “Zheng et al.”
Line 66: Species name is not in italics
Line 66: Please change “Zheng et al. reported that when the fish Carassius auratus were injected” to “Zheng et al. reported, in Carassius auratus, that when the fish were injected”
Line 67: Please change “were injected with 10 mg/kg body weight of each of nine and seventeen phthalates for several days” to “were injected with 17 different PAEs at a concentration of 10 mg/kg for several days”
Line 69: Please change “phthalates” to “PAEs”
Line 72-73: “Zebrafish are commonly used as a promising model organism for studies on the toxicity
of pollutants due to their high homology with vertebrates” This sentence needs some refences to be supported.
Line 76: “This information is intended to find useful biomarkers of DMP pollution.” I would move this to discussion.
Line 80: Please correct “grown” with “reared”
Line 80: How many fish?
Line 81: What about tank volume, water parameters and acclimation procedures?
Line 84: You should specify the local regulation that was followed
Line 86: “DMP for the exposure experiments was solubilized in acetone” This must be supported by a reference
Line 90: “laboratory system” please specify
Line 90: “the OECD” delete “the”
Line 91: please specify volume of experimental tanks
Line 93: “the solvent control” delete “the”
Line 93: “the test groups” which groups do you mean? Not clear
Line 94: why did you use 0.004% acetone in both controls, it should be better clarified and supported by citation. Should there not be blank control only in the system water?
Line 99: “After the LC50 values were determined” please change to “After LC50 value was determined”
Line 102: what about replicates and number of fish of control groups?
Line 103: How did you collect these 20 fish?
Line 103: “liver samples were dissected” how did you perform dissection? (Stereomicroscope? washes in PBS?) please be more specific.
Line 103: Information about pain-release procedures are missing, please specify anesthesia and euthanasia methods and add related references .
Line 104: “the MDA” please delete “the”
Line 105: Please choose one acronym (RT-PCR or qRT-PCR) and be consistent throughout the text.
Line 107: “The homogenates” delete “the”
Line 109: “The supernatants” delete “the” (This mistake is repeated many times above throughout the text, please correct)
Line 111: Please correct “analysed” with “analyzed” according to English US style that you already used throughout the text
Line 120: Please change “Catalase activity” to “CAT activity”
Line 122: Please change “Catalase activity” to “CAT activity”
Since you have used the acronym once you should not use the full form again: this error is often found in the manuscript, please check.
Line 131: Please delete “real-time polymerase chain reaction”
Line 146: Please change “of DMP was observed to” to “of DMP was seen to”
Line 154: “compared to the control” which one?
Line 171: Please correct “and GST activity increased.” to “and increased GST activity.”
Line 173: “than those of the control” You should write which control group or if you mean both please write “control groups”
Line 175: Please correct “SOD, CAT and GST” in uppercase
Line 178 - 182: same as line 175
Line 181: Please correct “and then the” to “followed by”
Line 191: “Significant differences from the control are indicated by an asterisk”. Is the asterisk missing from the figure or are there no significant differences?
Line 199: Same question as line 191
Line 214: Please correct “sod gene. (b) The cat gene. (c) The gst gene.” With “SOD gene expression. (b) CAT gene expression. (c) GST gene expression.”
Line 222: Please change “global pollutant that pollutes water resources and aquatic organisms” to “PAEs have become a global pollutant that contaminates water resources and affects aquatic organism health”
Line 225: “in the study” please correct to “in a study”
Line 226-227: Please add also the Latin name of species in brackets
Line 232 – 236: Please change “defence” to “defense”
Line 232 – 242: This is more an introduction about antioxidant enzymes not a discussion, please move or delete.
Line 234: delete “the well-known”
Line 240: “analysing” to “analyzing”
Line 242 – 244: This sentence is not well written, please clarify
Line 248: change “a trend of rising” to “a rising trend”
Line 250: “The activity of CAT” change to “The activity of CAT”
Line 269: Please correct “cause the mortality of the fish to increase” with “cause increased fish mortality.”
Author Response
Dear Reviewer:
I truly appreciate your very comprehensive comments, which are not only very valuable for this manuscript and my research but also inspiring my whole career guided by your rigorous spirit. Please see the attachment of my responses.
Once again thank you so much.
Best wishes
Chai Yingmei

Reviewer 3 Report
It is not clear in Figure 2 (asterisks not shown), nor in the text referring to the data in Figure 2 (lines 158-162 on page 5) whether or not there is a statistical difference.
Since the asterisks are not shown in Figure 3, the interpretation of the results shown is difficult.
Important information is missing in Table 1. For example, genbank accession number, amplicon size and reference of primers.
Can the gene that encodes beta-actin be used as a reference gene in gene expression studies in adult zebrafish liver?
Author Response
Dear Reviewer:
Thank you so much for your comments which are very helpful to this manuscript and my future work. My responses are as following:
Open Review
(x) I would not like to sign my review report
( ) I would like to sign my review report
English language and style
( ) Extensive editing of English language and style required
( ) Moderate English changes required
(x) English language and style are fine/minor spell check required
( ) I don't feel qualified to judge about the English language and style
Yes |
Can be improved |
Must be improved |
Not applicable |
|
Does the introduction provide sufficient background and include all relevant references? |
(x) |
( ) |
( ) |
( ) |
Is the research design appropriate? |
(x) |
( ) |
( ) |
( ) |
Are the methods adequately described? |
(x) |
( ) |
( ) |
( ) |
Are the results clearly presented? |
( ) |
(x) |
( ) |
( ) |
Are the conclusions supported by the results? |
(x) |
( ) |
( ) |
( ) |
Comments and Suggestions for Authors
It is not clear in Figure 2 (asterisks not shown), nor in the text referring to the data in Figure 2 (lines 158-162 on page 5) whether or not there is a statistical difference.
It has been revised.
Since the asterisks are not shown in Figure 3, the interpretation of the results shown is difficult.
It has been revised.
Important information is missing in Table 1. For example, genbank accession number, amplicon size and reference of primers.
Table 1 has been revised as following:
Primer |
Primer sequence (5’-3’) |
Annealing temperature (Tm) |
Product length (bp) |
Amplification efficiency |
SOD primer-F |
TCCGCACTTCAACCCTCA |
58.44 |
215 |
97.0% |
SOD primer-R |
CCTCATTGCCACCCTTCC |
57.66 |
||
CAT primer-F |
TACCAGTCAACTGCCCGTAC |
59.40 |
145 |
96.5% |
CAT primer-R |
GACTCAAGGAAGCGTGGC |
58.43 |
||
GST primer-F |
CCAACCACCTCAAATGCT |
55.09 |
150 |
98.1% |
GST primer-R |
ACGGGAAAGAGTCCAGACAG |
59.03 |
||
Beta-actin primer-F |
CGAGCAGGAGATGGGAACC |
59.86 |
214 |
99.5% |
Beta-actin primer-R |
CAACGGAAACGCTCATTGC |
58.29 |
The relevant gene sequences are as following:
Gene |
Species |
Accession |
SOD |
Danio rerio(SOD1) |
NM_131294.1 |
Danio rerio(SOD2) |
NM_199976.1 |
|
Oreochromis niloticus |
JF801727.1 |
|
Scophthalmus maximus |
MG253620.1 |
|
Labeo rohita |
KX650370.1 |
|
Paralichthys olivaceus |
EF681883.1 |
|
Seriola lalandi |
KT229634.1 |
|
Tachysurus fulvidraco |
KX455916.1 |
|
CAT |
Danio rerio |
AJ007505.1 |
Oreochromis niloticus |
JF801726.1 |
|
Scophthalmus maximus |
MG253621.1 |
|
Paralichthys olivaceus |
GQ229479.1 |
|
Siniperca chuatsi |
KJ578923.1 |
|
Oplegnathus fasciatus |
KT229635.1 |
|
Kryptolebias marmoratus |
NM_001329365.1 |
|
Labeo rohita |
KX650368.1 |
|
Seriola lalandi |
KT229635.1 |
|
GST |
Danio rerio |
NM_001045060.2 |
Danio rerio(Rho-class) |
BC139572.1 |
|
Oreochromis niloticus |
EU107284.1 |
|
Paralichthys olivaceus |
EU182592.1 |
|
catfish |
GH679923.1 |
Can the gene that encodes beta-actin be used as a reference gene in gene expression studies in adult zebrafish liver?
Beta actin-gene from zebrafish was identified as the most stably expressed reference genes to normalize the template. Many reports show that the beta-actin has been used as the reference gene on the influence of contaminants in zebrafish. (Such as Quintaneiro C, et al., 2019; Qian L, et al., 2018). It also used in the adult zebrafish ( Such as Zou SS, et. al., 2019). Chinese researchers have used this kind of gene as the reference in zebrafish liver and published the paper in Chinese. (Liu L, et al. 2015)
References:
Quintaneiro C, Teixeira B, Benedé JL, Chisvert A, Soares AMVM, Monteiro MS. Toxicity effects of the organic UV-filter 4-Methylbenzylidene camphor in zebrafish embryos.Chemosphere. 2019, 218:273-281.
Qian L, Cui F, Yang Y, Liu Y, Qi S, Wang C. Mechanisms of developmental toxicity in zebrafish embryos (Danio rerio) induced by boscalid. Sci Total Environ. 2018, 634:478-487.
Song-Song, Zou, Jing, et al. Thymosin participates in antimicrobial immunity in zebrafish.[J]. Fish & Shellfish Immunology, 2019. 87:371-378.
刘林, 赵群芬, 金凯星, et al. 纳米氧化锌对斑马鱼肝脏的毒性效应[J]. 环境科学, 2015, 36(10):3884-3891. Liu Lin, Zhao Qun-fen, Jin Kai-xing, et al. Toxic effect of Nano-ZnO in liver of Zerbrafish. Environmental Scinece, 2015(10): 3884-3891.
Once again really appreciate your comments. If you have other comments regarding my revision and response, please feel free to touch me.
Best wishes
Chai Yimei